# Preparation, Identification and Preliminary Application of the Fenvalerate Monoclonal Antibody in Six Kinds of Dark Tea

**DOI:** 10.3390/foods12051091

**Published:** 2023-03-03

**Authors:** Kang Wei, Qihui Yang, Yang Wei, Yuanfeng Wang, Naifeng Xu, Xinlin Wei

**Affiliations:** 1Department of Food Science & Technology, School of Agriculture and Biology, Shanghai Jiao Tong University, Shanghai 200240, China; 2Institute of Food Engineering, College of Life Science, Shanghai Normal University, 100 Guilin Road, Shanghai 200234, China

**Keywords:** hapten, hybridoma technique, ELISA, monoclonal antibody, test strip

## Abstract

Fenvalerate has the advantages of a broad insecticidal spectrum, high efficiency, low toxicity and low cost, and it is widely used in agriculture, especially in tea, resulting in the accumulation of fenvalerate residues in tea and the environment, posing a serious threat to human health. Therefore, the timely monitoring of fenvalerate residue dynamics is vital for ensuring the health of humans and the ecological environment, and it is necessary for establishing a fast, reliable, accurate and on-site method for detecting fenvalerate residues. Based on the methods of immunology, biochemistry and molecular biology, mammalian spleen cells, myeloma cells and mice were used as experimental materials to establish a rapid detection method of an enzyme-linked immunosorbent assay to detect the residues of fenvalerate in dark tea. Three cell lines—1B6, 2A11 and 5G2—that can stably secrete fenvalerate antibodies were obtained by McAb technology, and their sensitivities (IC_50_) were 36.6 ng/mL, 24.3 ng/mL and 21.7 ng/mL, respectively. The cross-reaction rates of the pyrethroid structural analogs were all below 0.6%. Six dark teas were used to detect the practical application of fenvalerate monoclonal antibodies. The sensitivity IC_50_ of the anti-fenvalerate McAb in PBS with 30% methanol is 29.12 ng/mL. Furthermore, a latex microsphere immunochromatographic test strip with an LOD of 10.0 ng/mL and an LDR of 18.9–357 ng/mL was preliminarily developed. A specific and sensitive monoclonal antibody for fenvalerate was successfully prepared and applied to detect fenvalerate in dark teas (Pu‘er tea, Liupao tea, Fu Brick tea, Qingzhuan tea, Enshi dark tea and selenium-enriched Enshi dark tea). A latex microsphere immunochromatographic test strip was developed for the preparation of rapid detection test strips of fenvalerate.

## 1. Introduction

Tea is consumed as the second largest beverage after water; it has been popular with people worldwide for a long time due to its special flavors. Furthermore, the beneficial functions, including antioxidant functions, hypoglycemic functions and the regulation of sleep as well as neurodegenerative diseases, have attracted much attention from researchers [1,2,3,4]. Nevertheless, the excessive use of various pesticide residues—for instance, the organophosphorus, organochlorine, carbamate, pyrethroid, organic nitrogen and other pesticides in tea—has been affecting the consumption and trade of tea and posing potential risks to health, even at a low level [5,6].

Fenvalerate (Fen) is a type II pyrethroid with the characteristic +α-cyano in its structure. Since Elliott et al. (1961) modified the structure of natural pyrethrin to develop permethrin, cypermethrin and deltamethrin, Fen has been subsequently developed by Ohno et al. (1976) and was commercialized in 1976 [7,8]. As a kind of pyrethroid, Fen has shown the characteristics of an efficient and broad-spectrum insecticidal, a low toxicity for mammals and a high stability, and it has been widely used to control pests [9,10]. Unfortunately, the large-scale use will inevitably pose a threat to the ecological environment and human beings [11]. Due to the wide application in the control of pests and diseases in vegetables and fruit trees, this kind of pyrethroid has entered into the human body through the food chain [12]. The long-term consumption of food with trace amounts of Fen residues exceeding the standard limit may cause related diseases of reproduction, neurological diseases, endocrine diseases and tumors in mammals [13,14]. It was reported that Fen could be absorbed by male mice, which caused the swelling and vacuolization of mitochondria in testicular spermatocytes. Meanwhile, the lysosomes were increased and the endoplasmic reticulum was greatly expanded [13]. Moniz et al. (2005) has found that the perinatal exposure to Fen resulted in abnormal hormone secretion, which interfered with brain organization in male pups and led to a delayed sexual maturation and a reduction in sexual behavior in female offspring [14]. Moreover, the exposure to Fen also increased the incidence of hepatocellular tumors, while the incubation period of lymphoma was shortened [15]. In order to ensure safety, many countries have strictly limited the maximum residue limit of Fen in agriculture products. For instance, the maximum residue limits for Fen in vegetables were regulated in the range of 0.02–0.2 mg/kg in the European Union [16], while the maximum residue limit for Fen in cereals and vegetables was regulated at 3.1 μg/kg in China.

Since the detriment of Fen is threatening human health, several detection methods of Fen were developed to prevent Fen from entering the human body. Zhu et al. (2019) synthesized a solid-phase microextraction (SPME) polyurethane-based film for pesticide extraction in chrysanthemum tea and used GC–EDC to detect Fen with a recovery rate of 97.5% and an LOD of 0.5 μg/kg [17]. Suman Gupta et al. (1996) used Fourier transform infrared (FTIR) spectroscopy for the determination of Fen in emulsifiable concentrate (EC) formulations, Fen was extracted with acetone and purified by thin-layer chromatography and spectroscopically; the carbonyl absorption band was determined at 1755 cm^−1^, and Fen was detected with a recovery rate of 93–99% and an LOD of 0.1 μg/kg [18]. In addition to chromatographic and spectral analysis methods, the sensor detection methods of Fen could recognize dependent variables associated with Fen, including enzymes, antibodies, molecularly imprinted polymers, aptamers and host–host recognizers based on fluorescence (FL), Raman spectroscopy and chemiluminescence signal changes [19]. Current sensors for the detection of Fen include capacitive chemical sensors, fluorescent sensors, SERS sensors, chemiluminescent sensors and immunosensors [20,21,22].

Although a variety of detection methods have been developed for Fen, none of these methods can meet the requirements of the rapid on-site detection of samples. Immunological methods, a new type of rapid detection technology, can accurately detect target analytes in samples containing complex matrix components, and they have been widely used in pesticide residue detection [23]. Presently, the widely used ELISA methods are the immobilized antigen competitive immunoassay (called the indirect competitive enzyme-linked immunosorbent assay, or ic-ELISA for short), immobilized antibody competitive immunoassay (direct competitive enzyme-linked immunosorbent assay, dc-ELISA) and double antibody sandwich method [24]. Song et al. (2011) used anti-Fen polyclonal antibody and dc-ELISA to detect Fen in tea samples; the recovery was 76.67~91.43% and the LOD was 0.16 mg/kg [25]. To the best of our knowledge, there has been no commercialized monoclonal antibody for the specific detection of Fen residues. It was reported that, different from vegetables, which are allowed to contain 0.05 to 10.0 mg/kg of Fen, Fen is strictly forbidden in tea in China. The Chinese Ministry of Agriculture stipulated that the use of Fen pesticides on tea trees was prohibited in 1999 to ensure the export of tea [26]. Nevertheless, a previous study detected Fen residues from one hundred and one tea samples from China, including green tea, dark tea, scented tea, black tea and oolong tea, and reported that there was a 24.8% detection ratio of Fen from these tea samples [27]. The remaining Fen in tea is threatening the health of tea consumers; therefore, rapid and accurate of ELISA detection methods should be established to detect Fen residues in tea to regulate the tea production and market.

In this study, the haptens of Fen were synthesized and identified by LC–MS. The synthesis of complete antigens was optimized by MA and EDC methods and used for mice immunity. The anti-Fen monoclonal antibody was prepared by a hybridoma technique, and an ELISA method was established to detect Fen based on the screened McAbs. Six kinds of dark teas (Pu‘er tea, Liupao tea, Fu Brick tea, Qingzhuan tea, Enshi dark tea, selenium-enriched Enshi dark tea), which are deeply loved by the people of northwest China, were used to evaluate the practical application of the ELISA detection method of Fen. Further, a latex microsphere immunochromatographic test strip was developed to achieve the rapid, accurate and on-site detection of Fen. This study will provide a theoretical and application basis for the rapid detection of Fen residues.

## 2. Materials and Methods

### 2.1. Chemicals

Six-week-old female Bagg Albino (BALB/c) mice were purchased from SLAC Laboratory Animal Company (Shanghai, China). Mouse myeloma cells (SP2/0) were obtained from the National Collection of Authenticated Cell Cultures of China (Shanghai, China). Carrier proteins of keyhole limpet hemocyanin (KLH) and N-hydro-xysuccinimide (NHS) were obtained from Solarbio Biotechnology Company (Beijing, China). 1-(3-dimethylaminopropyl)-3-ethylcarbodiimide hydrochloride (EDC) was obtained from Sigma Aldrich (St. Louis, MO, USA). Fenvalerate (analytical standard) was purchased from Dr. Ehrenstorfer GmbH (Augsburg, Germany). All other chemicals were of analytical grade, unless otherwise specified.

### 2.2. Synthesis and Identification of Fenvalerate Haptens

The linking arm and carboxyl group were introduced at the α-cyano group in the Fen structure as a hapten; the synthetic procedure of Fen haptan is shown in Appendix A, according to the previous study [28]. Then, the Fen hapten was identified by the liquid chromatography–mass spectrometry (LC–MS) analysis according to the method reported by Martínez et al. (2006) [29]. Briefly, the LC–MS conditions were performed with the Acquity UPLC I-class/VION IMS QTOF (waters, Milford, MA, USA) according to Martínez, with a slight modification: solvent A was set as acetonitrile and solvent B was set as ammonium formate 50 mM and 5% acetonitrile, acidified at pH 3.5 by adding formic acid to perform the program (30% B in 0–3 min, 30–20% B in 4–8 min, 20–0% B in 8–10 min, 0% B in 10–13 min). The haptens were identified according to the liquid chromatographic peak and mass spectral molecular weight.

### 2.3. Synthesis of Immunogens

The synthesized hapten was successfully coupled with the carrier proteins BSA, KLH and OVA in different molar ratios to synthesize the complete antigen of Fen using the mixed anhydride method (MA method) and EDC method [30].

#### 2.3.1. MA Method

Carrier protein BSA: Solution A: 5.23 mg (0.01 mM) of the hapten was dissolved in 1 mL of N-N dimethylformamide (DMF) solution. Then, 13.1 μL of isobutyl chloroformate and 7.2 μL of tri-n-butylamine were added to the hapten solution, respectively. The reaction was activated under an ice bath and stirring conditions for 2 h. Solution B: According to the molar ratio of Fen:BSA = 30:1, 60:1 and 90:1, BSA was weighted and dissolved in 0.01 M PBS. After solution A was slowly added dropwise to solution B under the stirring conditions of an ice bath, the mixed solution was continuously stirred for another 4 h in an ice bath. Thereafter, the mixed solution was transferred to an activated dialysis bag. Dialysis was performed at 4 °C for 72 h with 0.01 M PBS, and the dialysate was replaced every 4–6 h. Finally, the antigen solution was dispensed into 1.5 mL centrifuge tubes, sealed with parafilm and frozen at −20 °C for future use.

Carrier Protein KLH: The carrier protein KLH was coupled with hapten by the MA method at a molar ratio (Fen: KLH) of 3000:1. The experimental procedure was the same as the carrier protein BSA-MA method.

#### 2.3.2. EDC Method

Carrier Protein BSA: Solution A: 5.27 mg (0.01 mM) of hapten was dissolved in 1 mL of DMF and then removed and placed into a centrifuge tube of 2.9 mg (0.015 mM) EDC. After dissolving, it was removed and placed into a centrifuge tube of 1.7 mg (0.015 mM) NHS. A total of 0.5 mL of DMF was added to the centrifuge tube containing the hapten to collect the residual hapten, and then the residual amount of EDC and NHS was collected in turn. The three solutions of hapten, EDC and NHS were combined, and the reaction was activated for 4 h under the condition of stirring in an ice bath. Solution B: BSA, according to the molar ratio of Fen:BSA = 30:1, 60:1 and 90:1, was weighted and dissolved in 0.01 M PBS. Under the stirring condition of an ice bath, solution A was slowly added dropwise to solution B, and the reaction was continued for 4 h. After the reaction, the solution was transferred to an activated dialysis bag. Dialysis was performed at 4 °C for 72 h with 0.01 M PBS, and the dialysate was replaced every 4–6 h. Finally, the antigen solution was aliquoted into 1.5 mL centrifuge tubes, sealed with parafilm and frozen at −20 °C for future use.

Carrier Protein KLH: The carrier protein KLH was coupled with hapten by the EDC method at a molar ratio (Fen: KLH) of 3000:1. The experimental procedure was the same as the carrier protein BSA-EDC method.

### 2.4. Synthesis of Coating Antigen

The coating antigen was synthesized by the MA and EDC method [30] as follows.

#### 2.4.1. MA Method

Carrier protein OVA: The carrier protein OVA was coupled with hapten by the MA method at a molar ratio (Fen: OVA) of 30:1. The experimental procedure was the same as the carrier protein BSA-MA method.

#### 2.4.2. EDC Method

Carrier protein OVA: The carrier protein OVA was coupled with hapten by the EDC method at a molar ratio (Fen: OVA) of 30:1. The experimental procedure was the same as the carrier protein BSA-EDC method.

### 2.5. Identification of the Complete Antigen of Fenvalerate

The methods of ultraviolet (UV) spectra and sodium dodecyl sulfate polyacrylamide gel electrophoresis (SDS-PAGE) were performed to identify whether the Fen complete antigen was synthesized successfully [31].

### 2.6. Preparation and Evaluation of Anti-Fenvalerate Monoclonal Antibody (McAb)

Mouse immunization was performed, and the process referred to the method of Xu et al. [30]. Then, the mouse antiserum that showed the highest titer (Appendix A) and the lowest IC_50_ value for approximately 0.8 μg/mL was chosen for the cell fusion and McAb preparation. The methods of cell fusion and screening were performed according to the classical procedure [32]. The hybridoma cells were identified by the ic-ELISA method with the standard of Fen after each subcloning; thereafter, the hybridoma cells with a high sensitivity and specificity were selected for subsequent testing.

The ascites were induced by the intraperitoneal injection of selected hybridoma cells into BALB/C mice to achieve the mass production of the monoclonal antibody. Subsequently, the McAb in ascites was pacificated by the caprylic acid–ammonium sulfate precipitation method [33].

The concentration of the monoclonal antibody was determined by the BCA kit (P0010, Beyotime, Shanghai, China), the purity of the monoclonal antibody was determined by SDS-PAGE gel electrophoresis, the titer of the monoclonal antibody was determined by indirect non-competitive ELISA and the affinity of McAb was determined by ic-ELISA [34]. The sensitivity of McAb is expressed as a half inhibitory concentration (IC_50_) with the following conditions [28]. Briefly, the coated antigen (Fen-EDC-BSA60) was diluted to 0.063 μg/mL, and the monoclonal antibodies obtained from the selected 1B6, 2A11 and 5G2 cell lines were diluted 243,000 times, 81,000 times and 81,000 times, respectively. The series of the concentration of the Fen standard was set as 0 ng/mL, 1 ng/mL, 3 ng/mL, 10 ng/mL, 30 ng/mL, 100 ng/mL, 300 ng/mL and 1000 ng/mL, and the logarithm value (lg) of the concentration of the Fen standard was taken as the X-axis, with OD450nm as the Y-axis. The standard curve was drawn in Origin8.5 (Microcal, Northampton, MA, USA) software, according to which the IC_50_ of the antibody could be obtained.

The cross-reactivities with nine permethrin pesticides (deltamethrin, permethrin, cypermethrin, bifenthrin, fenpermethrin, fluobenzene, beta-cyhalothrin, dextrofenthrin, cypermethrin) were obtained, and McAb was determined to clarify the specificity of the monoclonal antibody prepared [28]. The IC_50_ of the nine permethrin pesticides was measured by the ic-ELISA with the obtained McAb and the coating antigen [28]. The cross-reactivity was calculated with the following Formula (1):Cross-reactivity (CR, %) = IC_50_ of Fen/IC_50_ of Fen analogs × 100%(1)

### 2.7. Application of Fenvalerate Enzyme-Linked Immunosorbent Assay

The fenvalerate detection of dark tea using the enzyme-linked immunosorbent assay was performed according to the previous study, with some modifications [25]. A total of 1 g of dark tea powder was added into 10 mL of methanol and extracted for 30 min on a shaker. The supernatant was filtered by a 0.22 μm organic filter membrane. The tea extracts were diluted with a concentration sequence (1:50, 1:100, 1:200, 1:500, 1:1000, *v*/*v*) by the standard diluent (0.01 M PBS containing 30% methanol) to obtain the corresponding tea soup. The diluted tea extracts were used as a diluent for the Fen standard curve to optimize the dilution ratio of tea; thereafter, the influence of the tea substrate on the Fen measurement was eliminated.

The recovery percentage of the standard samples was measured with the following method. Briefly, PBS with 30% methanol and the optimized tea diluent were used, respectively, to dilute the Fen standard to obtain the standard curve. The actual amount of Fen detection was calculated by the standard curve, and the recovery rate of Fen was obtained by the following Formula (2):Recovery rate (%) = actual detected amount/added amount × 100%(2)

### 2.8. Preparation and Evaluation of Fenvalerate Immunochromatographic Strip

The fenvalerate immunochromatographic strip was prepared according to our previous study [35]. Briefly, 1.2 μg of 1B6 McAb was added into 50 μL of 300 nm red carboxyl microspheres, which were obtained from the VDObiotech company (Suzhou, China) and characterized in our previous study; after 10 min of vortex oscillation, the detection probe was synthesized for further use [35].

The optimized 0.34 mg/mL coating antigen and 1.0 mg/mL goat anti-mouse were used in the T line and C line of the latex microsphere immunochromatographic test strips for the Fen detection, respectively. The sensitivity of the test strip was determined. Briefly, the Fen was diluted for a sequence of concentrations of 0, 10, 30, 50, 100, 300, 500 and 1000 ng/mL; thereafter, the Fen was mixed with the produced probe for 3 min. The mixed solution was added onto the sample pad for a 5 min reaction; then, the color changes of the T line and the C line were observed and recorded by Image J (Version 1.8.0).

### 2.9. Statistical Analysis

All the experimental data are presented as the mean ± standard deviation (SD); the figure was generated using Origin8.5 software (Microcal, Northampton, MA, USA).

## 3. Results

### 3.1. Identification of Fenvalerate Hapten and Complete Antigen

Based on the method of Jiang et al. (2010) [28], the hapten of Fen was synthesized, and the characterization of the Fen hanpten was performed by LC–MS. As depicted in Figure 1A, there was a strong electrical signal between 6.7 min and 7.2 min without miscellaneous peaks, indicating the successful synthesis of a pure compound. Furthermore, mass spectrometry (MS) was used to identify the two signal peaks, and the corresponding results of the two peaks were almost identical in Figure 1C,D. The MS analysis was used to further verify the synthesis results, and it was evident that the molecular weight of the target Fen hapten was 508.15 Da, while the molecular weight of the target Fen hapten was 509.16 Da. Therefore, Fen hapten was synthesized successfully with a high purity, which could be further used to prepare the Fen complete antigen.

The EDC method and MA method are the classic methods of antigen synthesis described by Watanabe et al. (1999) and Singh (1978), respectively [36,37]. It was reported that Xu et al. (2015) used these method to successfully synthesize the McAb of chloramphenicol [30]. Therefore, the EDC method and MA method were performed to optimize the synthesis of the antigens. Since the hapten and common carrier proteins including BSA, KLH and OVA all exhibit the special UV absorption peaks, respectively, the UV absorption peaks will be altered after the combination of the hapten and carrier proteins. Therefore, UV spectrum characterization was performed to detect whether hapten was successfully coupled with the carrier proteins. Since the scanning results of the hapten, the carrier proteins and their binding compounds in the wavelength range of 200–800 nm showed that the wavelength range corresponding to the maximum absorption peak was 250–320 nm, UV absorption peaks in the wavelength range of 250–320 nm were selected for further analysis. As seen in Figure 2, hapten coupled with BSA in both the MA and EDC methods showed a similar waveform to that of BSA and a similar maximum absorption peak wavelength to that of hapten, which demonstrated that hapten was successfully conjugated to the carrier protein of BSA. In terms of the carrier protein of KLH and OVA, the complete antigen synthesized by the MA and EDC methods had the characteristics of hapten and carrier proteins, indicating the successful synthesis of the complete antigen.

Owing to the combination of carrier proteins, the molecular weight of the complete antigen detected by SDS-PAGE will be greater than that of any single carrier protein. Since the molecular weight of KLH was too large to determine, the rest of the synthesized antigen was detected by SDS-PAGE (Figure 3) [35]. The bands of five kinds of immunogens (Fen-MA-BSA30, Fen-MA-BSA60, Fen-MA-BSA60, Fen-EDC-BSA30 and Fen-EDC-BSA60) and two kinds of coating antigens (Fen-MA-OVA30 and Fen-EDC-OVA30) all showed an obvious upward shift of the bands, indicating that the synthesized antigens have a larger molecular weight than the corresponding carrier proteins (Figure 3), and the complete antigens were successfully synthesized. Notably, as the proportion of hapten in the conjugate increased, the upward shift of the band was more obvious; meanwhile, the upward shift of the band of the complete antigen synthesized by the EDC method was more obvious than that synthesized by the MA method.

### 3.2. Animal Immunity and Screening of the Positive Cell

Fen-MA-OVA30 and Fen-EDC-OVA30 were detected with the ic-ELISA method, and Fen-EDC-OVA30 was chosen, with a better effect in the concentration of 0.38 μg/mL, for further use. Thereafter, seven kinds of immunogens, including Fen-MA-BSA30, Fen-MA-BSA60, Fen-MA-BSA90, Fen-EDC-BSA30, Fen-EDC-BSA60, Fen-MA-KLH3000 and Fen-EDC-KLH3000, were injected into BALB/c mice for six immunizations. The immunogen synthesized by the EDC method showed a higher titer and lower IC_50_ than the MA method. Additionally, the best antigen among the seven kinds of antigens was synthesized by the EDC method, with a 60:1 mole ratio of hapten to BSA. Therefore, the splenocyte induced by the immunogen Fen-EDC-BSA60 was chosen for further experiments. After the cell fusion, the cell lines with a high sensitivity and strong specificity should be screened by ic-ELISA. Three cell lines with stable anti-Fen antibody secretion, 1B6, 3A11 and 5G2, were obtained after four subclonal screenings. Notably, the results of the third and fourth subclone showed a 100% positive rate of the three cells, indicating that the cells were able to stably secrete antibodies.

### 3.3. Characterization of McAb against Fenvalerate

Due to the interference of various miscellaneous proteins in the production of monoclonal antibodies, the performance of the antibody will be impaired, which indicates that ascites purification is necessary for McAb production. Saturated ammonium sulfate was selected for the purification of McAb due to its most widely used mild and effective characteristics. As shown in Figure 4, the McAb subtype of the three cell lines was IgG1, and the light chain type was Kappa type.

The concentrations of McAb secreted by 1B6, 2A11 and 5G2 were 8.75 mg/mL, 6.82 mg/mL and 6.08 mg/mL, respectively, indicating the high yield of McAbs in the three cell lines. To ensure the stability and accuracy of the subsequent experiments, SDS-PAGE was performed to identify the purity of the purified antibody. The bands of proteins in Figure 4 showed that the 1B6, 2A11 and 5G2 cell lines secreted the antibodies with clear heavy-chain and light-chain bands. The molecular weight of the heavy chain was about 50 kDa, and the molecular weight of the light chain was about 25 kDa, indicating a high purity of the antibodies. The antibody titer is one of the important indicators of the antibody level. As shown in Appendix A, all the McAbs showed a high titer to 2.19 × 10^6^, which demonstrated that all three fused cells could produce sufficient target antibodies with a high purity. Subsequently, the affinity of McAb was determined by ic-ELISA to explore the binding strength of McAb to antigen.

The affinity constant (Ka) between 10^7^ and 10^12^ L/mol indicates the strong affinity of the antibody. According to the affinity assay method and operation procedure, the affinity measurement curve is shown in Figure 5. The Ka values of the 1B6, 2A11 and 5G2 cell lines were 2.247 × 10^9^ L/mol, 1.94 × 10^8^ L/mol and 4.52 × 10^8^ L/mol, respectively, indicating that the affinity of the three strains was sufficiently high. The sensitivity, expressed by the half inhibition rate (IC_50_), is the key indicator for evaluating the performance of McAb. In the current work, an ic-ELISA method was introduced to determine the IC_50_ of McAb [28]. As shown in Figure 6 and Table 1, the IC_50_ of the McAb secreted by 1B6 was 36.6 ng/mL, and the linear detection range (LDR) was 12–106 ng/mL. The IC_50_ of the McAb secreted by 2A11 was 24.29 ng/mL, and the LDR was 5.9–99 ng/mL. The IC_50_ of 5G2 was 21.65 ng/mL, and the LDR was 8.5–54.8 ng/mL. The higher titers and larger LDR of the McAb secreted by 1B6 among the three McAbs were chosen for further study.

In order to clarify the specificity of McAb against Fen, the cross-reactivity to nine permethrin pesticides (deltamethrin, permethrin, cypermethrin, bifenthrin, fenpermethrin, chlorothrin, beta-permethrin, beta-fluorothrin, beta-permethrin, dexfenthrin, and cypermethrin) which were analogs of Fen were determined by ic-ELISA. As demonstrated in Table 2, three McAbs were able to cross-react with fenvalerate, specifically, while the cross-reactivity of the remaining eight permethrin pesticides was less than 1%, which was far lower than that of antiserum to Fen, reported by Song et al. (2011), and that of McAb, prepared by Jiang et al. (2010) [25,28], indicating the great specificity of the produced McAbs against Fen.

### 3.4. Application of McAb in the Fenvalerate Detection of Dark Tea

Dark tea, as one of the six kinds of teas in China, is popular with people because of its function and flavor [38,39]. Nevertheless, the Fen exceeding the detection standard in teas could pose potential risks to tea consumers. In this study, Fen detection has been performed on six kinds of dark tea to evaluate the application of the McAb in the Fen detection of dark tea. With the addition of the Fen standard into Pu‘er tea, Liupao tea, Fu Brick tea, Qingzhuan tea, Enshi dark tea and Se-enriched Enshi dark tea, the experiment was carried out to evaluate the practicability of the established ELISA method. The curves of the standard diluent and the tea substrate diluent were performed to eliminate the potential interference of the tea substrate to the ELISA detection system. As seen from Figure 7, compared with the standard curve of PBS containing 30% methanol, the influence of the tea substrate on ELISA was eliminated, while tea substrates were diluted 100 times with PBS containing 30% methanol. When the added amount of the Fen standard was 25 ng/mL, 50 ng/mL and 100 ng/mL, the recovery of Fen in the dark teas was 73.2–88.8%, 90.2–96.6% and 96.7–99.8%, respectively (Table 3). Compared with the ELISA method developed by this study, the method from Song et al. (2011) showed that the recovery of Fen in tea detection ranged from 76.67 to 91.43%, whereas the spike level was set from 60 μg/mL to 700 μg/mL, which was much higher than the 25–100 ng/mL determined in this study, indicating the suitability and high sensitivity of the established ELISA method based on the produced McAbs in this study for applications in tea detection [25].

### 3.5. Sensitivity Evaluation of the Latex Microsphere Immunochromatographic Test Strip

In order to improve the shortcomings, including the limited test site, high cost and complex sample pretreatment of the existing Fen detection methods, a latex microsphere immunochromatographic test strip was developed based on the ELISA method. As shown in Figure 8, the log_10_ value of the Fen standard solution concentration was used as the X-axis, the corresponding T/C value measured by ImageJ was used as the Y-axis, the sensitivity standard curve of the test strip was obtained, the limit of detection (LOD) of the test strip was 10.0 ng/mL, the LDR was 18.9–357 ng/mL and the IC_50_ was 82.1 ng/mL.

## 4. Discussion

Fenvalerate, as a typical type II pyrethroid, has been widely used in agriculture and crop control, resulting in the continuous accumulation of Fen residues in the crop and environment, which eventually enter the human body through the food chain, posing a safety hazard to human health [40]. With the continuous development of detection technology, the application of GC–MS, spectroscopy, sensors and the immunoassay has been performed to develop various Fen detection methods [17,41,42]. The accuracy and specificity of classical detection methods are usually insufficient [25,43]. Meanwhile, though newly developed detection methods including spectral analysis and sensor methods could meet the requirements of high sensitivity and a low detection limit, the expensive devices, the lack of portability, the time-consuming characteristics and the laborious nature limited their wide application [42]. Immunological methods can accurately detect target analytes in samples consisting of complex components [23]. Furthermore, immunological methods are the basis for the development of immunochromatographic strips, which could achieve the rapid on-site detection of samples. Scholars have focused on the ELISA method for detecting fenvalerate [25,28].

The establishment of immunological methods was related to the production of specific McAb, while McAb production begins with the synthesis of haptens. Therefore, the Fen hapten was synthesized and evaluated in this study. Interestingly, two peaks were shown in the liquid chromatogram of Figure 1, which may be related to the structure of Fen. It was reported that there are two chiral carbon atoms in the Fen structure, leading to the existence of four Fen optical isomers. However, four Fen optical isomers could be divided into two groups of enantiomers, which were usually separated into two peaks by chromatographic analysis [44,45]. Further MS analysis proved that the two peaks corresponded to the same substance of the target Fen hapten. It was reported that a method based on hybridoma technology has been established to produce the McAb against Fen [28]. On this basis, the MA and EDC methods were used to couple the hapten to the common carrier protein to optimize the process of the complete antigen synthesis of Fen; thereafter, seven immunogens were optimized to produce antiserum [30,36,37]. Notably, this study was consistent with the study of Jiang et al. (2010), which found that BSA was the more suitable protein for coupling with the hapten of Fen [28]. The complete antigen Fen-EDC-BSA60 was optimized to be used as both an immunogen and coating antigen. In comparison with the McAb produced by Jang et al. (2010), the IC_50_ of the McAb secreted by the synthesized hybridoma cell was as low as 21.65 ng/mL, which achieved a great enhancement in the sensitivity of anti-Fen McAb [28]. The specificity of McAb was the key factor in evaluating the function of McAb. Although the IC_50_ of the monoclonal antibody against Fen prepared by Wang et al. (2011) was as low as 2.2 μg/kg, the McAb could also recognize the other five pyrethroids, which cannot be applied in the actual test of fenvalerate [46]. In the present study, the specificity of the McAbs was detected, and the CR% of synthesized McAbs with other 9 Fen analogs was lower than 1%, indicting the great specificity of McAbs. Moreover, the McAb was applied against Fen for the detection of dark tea with the ELISA method. It was reported that dark tea showed a special flavor and several activities that could improve the health of the human body [38,39]. Compared with the competitive direct ELISA method based on the polyclonal antibody against Fen used in the detection of tea, which showed an 81.43% recovery and a spike level of 100 μg/mL, the high recovery rate (96.7–99.8%) in a lower spike level (100 ng/mL) and the low coefficient of variation of the McAbs prepared in this study indicated the McAbs could be used in the detection of Fen in dark tea [25]. Although the Chinese Ministry of Agriculture forbade the use of fenvalerate in tea in 1999, research involving the fenvalerate detection of 101 kinds of tea samples found that there are still excessive Fen residues [27]. It was necessary to prepare a rapid and on-site method for detecting the fenvalerate. In the present study, a latex microsphere immunochromatographic test strip which could accomplish on-site Fen detection was successfully designed based on the ELISA method. The properties of the McAb, coating antigen and latex microspheres are the key components of the strip. It was reported that the optimal coating antigens and McAb could ensure the specificity and sensitivity of the strip, while the uniform latex microspheres could guarantee the stability of the strip [35]. Although the IC_50_ value of the McAb secreted by 1B6 was lower than that secreted by 2A11 and 5G2, 1B6 was selected to prepare the test strip considering the greater affinity, titer and linear range of the McAb secreted by 1B6. Under the optimal condition, the T/C value of the test strip decreased with the increase in the Fen concentration, which was consistent with the time-resolved fluorescent microspheres Eu lateral flow test strip developed by Zhang et al. (2022) [16]. The LOD of the test strip in Fen detection was calculated as 8 ng/mL, which was lower than the 51 ng/mL of the test strip developed by Zhang et al. (2022) [16]. Moreover, excessive Fen also could be determined by the visible color reaction of the latex microsphere immunochromatographic test strip, without any instrumental assistance, while the time-resolved fluorescent microspheres Eu lateral flow test strip should be equipped with a uviol lamp for Fen detection. It was obvious that the development of the latex microsphere immunochromatographic test strip based on anti-Fen McAb could achieve an enhancement in the portability and sensitivity of Fen detection, which provided the theoretical and application basis for the rapid detection of Fen residues in tea and other vegetable products.

## 5. Conclusions

In this study, the fenvalerate antigen synthesis method was optimized; mammalian spleen cells, myeloma cells and mice were used as experimental materials to prepare three kinds of anti-Fen monoclonal antibodies with a higher affinity and sensitivity based on the hybridoma technique. Immunology, biochemistry and molecular biology methods were combined to establish an indirect competitive enzyme-linked immunosorbent assay (ic-ELISA) for the determination of Fen residue in dark tea. To the best of our knowledge, there has been no available latex microsphere immunochromatographic test strip used in Fen detection. We prepared a novel latex microsphere immunochromatographic test strip with the McAb of a greater affinity, titer and LDR. The LOD of the test strip was 10 ng/mL, and the LDR was 18.9–357 ng/mL in the detection of Fen. Therefore, the prepared tool of the test strip will provide a potential application for the convenient and rapid on-site detection of Fen in tea and vegetable products. In the future, the sensitivity of the test strip in teas, vegetables and environment samples should be further evaluated. Furthermore, the nanomaterials coupled with the antibody in the strip, such as classical colloidal gold materials and advanced gold nanoflowers, will be applied to the assembly and performance test of the strip to screen out the optimal strip for fenvalerte detection, which will play an increasingly important role in environmental protection and food safety for maintaining human health.

## Figures and Tables

**Figure 1 foods-12-01091-f001:**
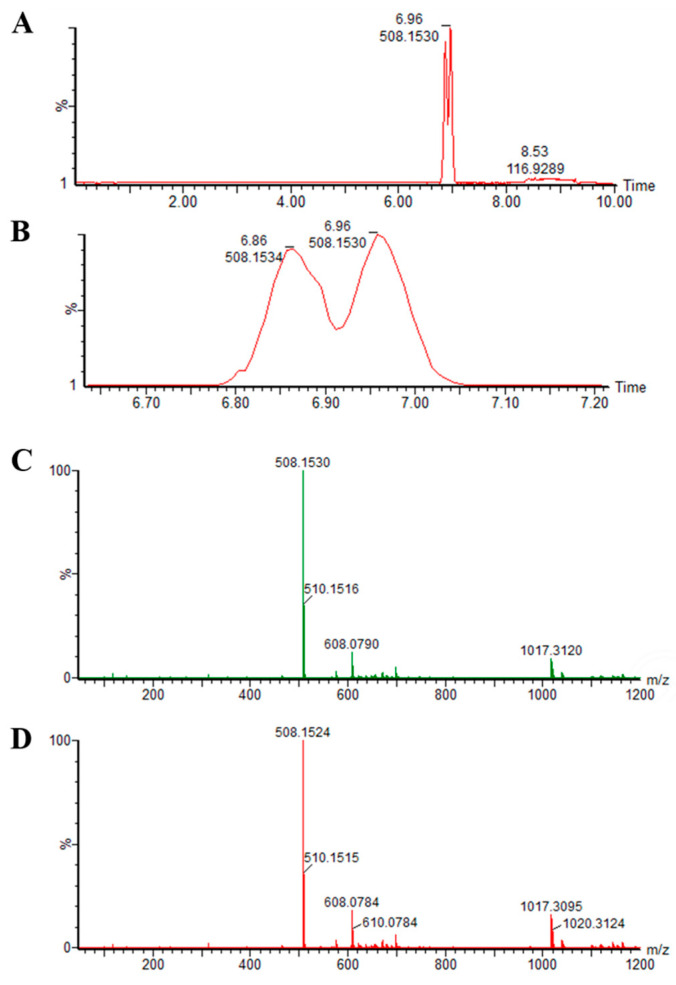
LC–MS analysis of synthesized Fen hapten. (**A**) Complete liquid chromatogram of synthesized Fen hapten. (**B**) The liquid chromatogram of synthesized Fen hapten during the retention time of 6.7–7.2 min. (**C**) Mass spectrogram of the peak with a retention time of 6.86 min. (**D**) Mass spectrogram of the peak with a retention time of 6.96 min.

**Figure 2 foods-12-01091-f002:**
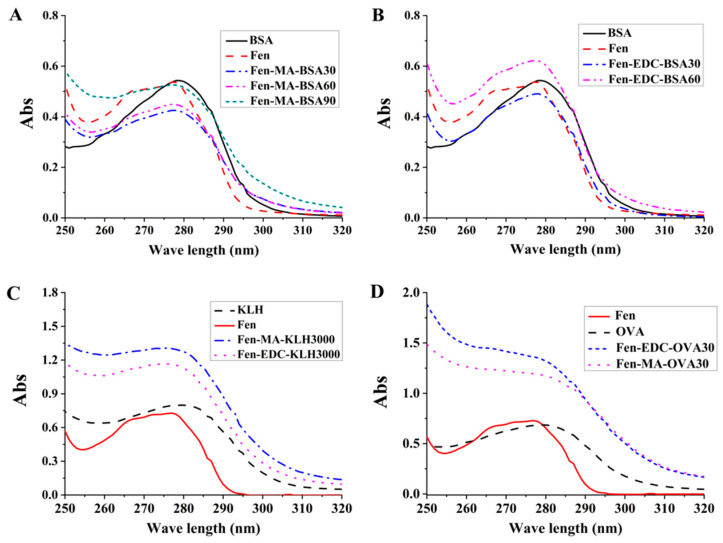
UV identification of synthesized complete antigens of Fen. (**A**) The complete antigen in which hapten was coupled to BSA by the MA method. (**B**) The complete antigen in which hapten was coupled to BSA by the EDC method. The antigen of Fen-EDC-BSA90 precipitated after dialysis and was not further studied. (**C**) The complete antigen in which hapten was coupled to KLH. (**D**) The complete antigen in which hapten was coupled to OVA.

**Figure 3 foods-12-01091-f003:**
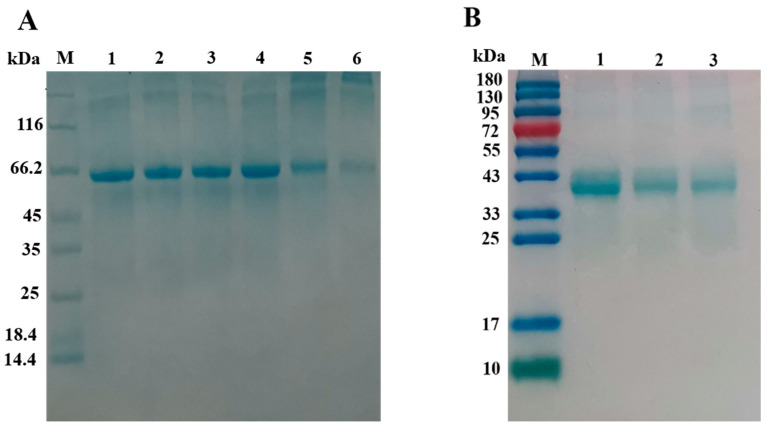
SDS-PAGE identification of synthesized complete antigens of Fen. (**A**) The complete antigen in which hapten was coupled to BSA. (**B**) The complete antigen in which hapten was coupled to OVA. In Figure 1A, M is the marker, 1 is BSA, 2 is Fen-MA-BSA30, 3 is Fen-MA-BSA60, 4 is Fen-MA-BSA90, 5 is Fen-EDC-BSA30 and 6 is Fen-EDC-BSA60. In Figure 1B, M is the marker, 1 is OVA, 2 is Fen-MA-OVA30 and 3 is Fen-EDC-OVA30.

**Figure 4 foods-12-01091-f004:**
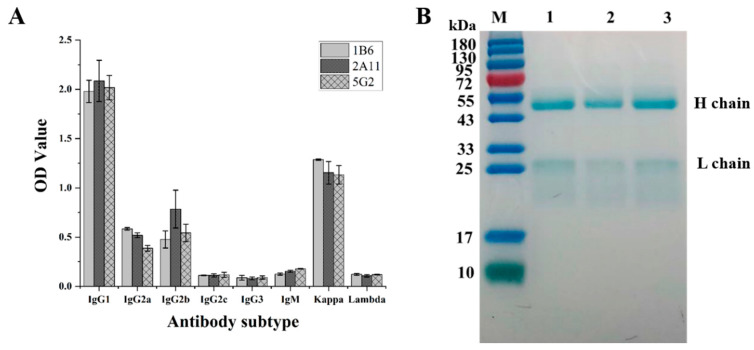
Identification of positive hybridoma cells. (**A**) Identification of the antibody isotype secreted by the selected cell lines. (**B**) Identification of the purity of the antibodies secreted by the selected hybridoma cells using the SDS-PAGE. In Figure 4B, M is the marker, 1 is the McAb secreted by 1B6, 2 is the McAb secreted by 2A11 and 3 is the McAb secreted by 5G2.

**Figure 5 foods-12-01091-f005:**
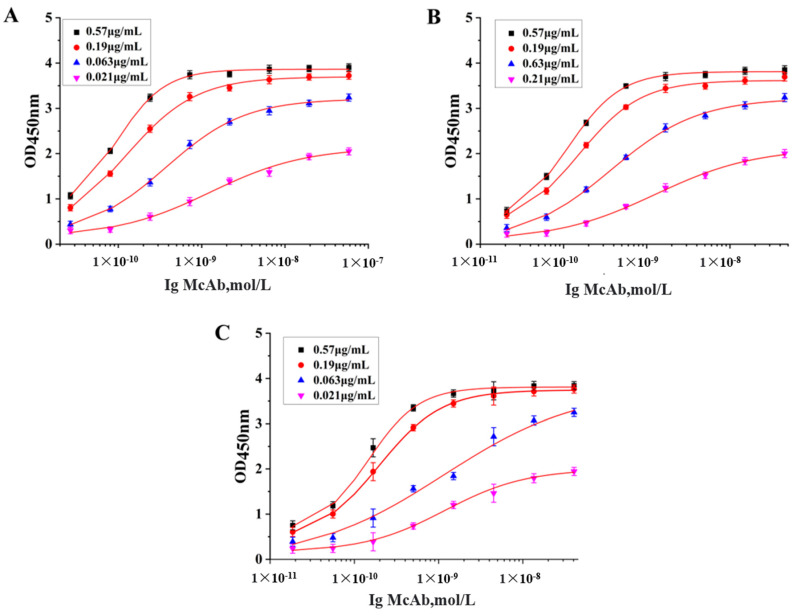
Identification of the affinity of the monoclonal antibody. (**A**) The affinity standard curve of McAb secreted by (**A**) 1B6, (**B**) 2A11 and (**C**) 5G2. Different-colored standard curves represent different concentrations of the coating antigen.

**Figure 6 foods-12-01091-f006:**
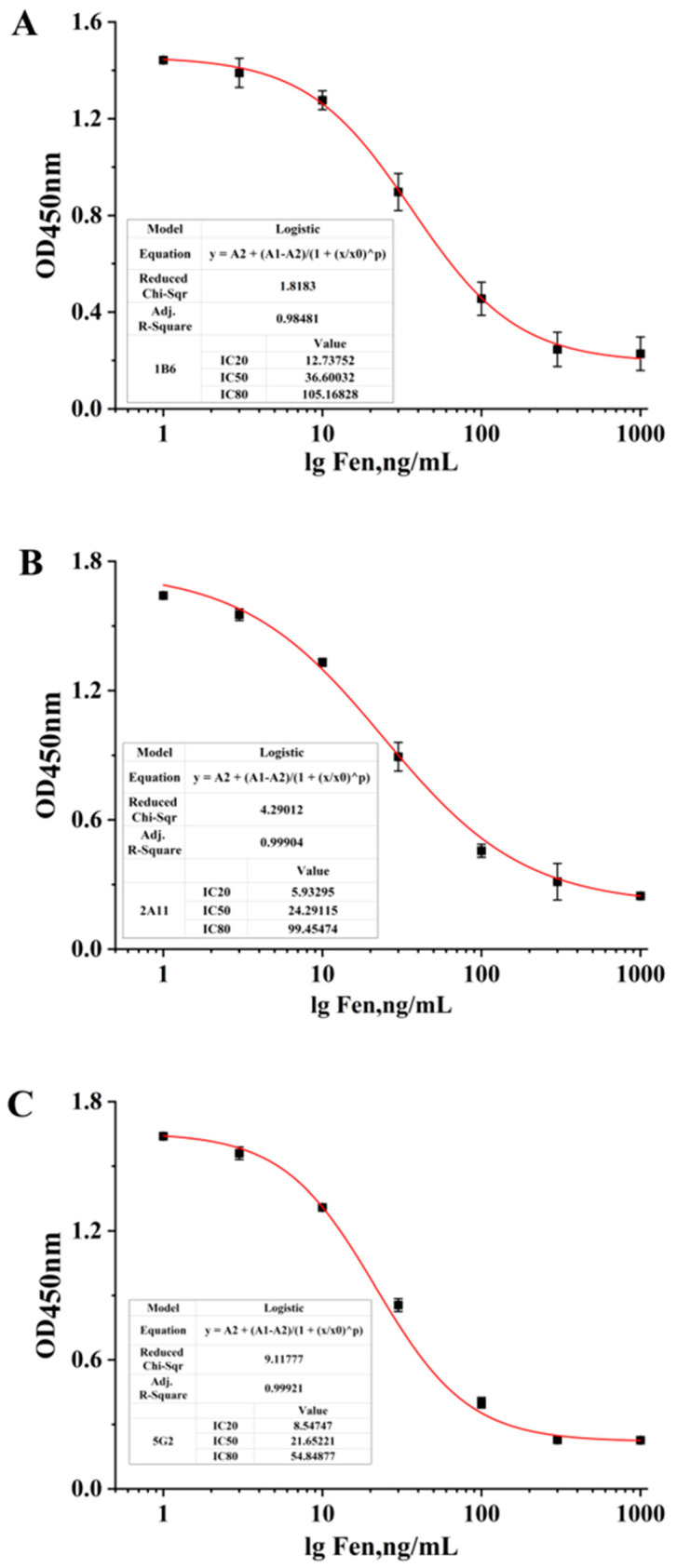
Identification of the sensitivity of the monoclonal antibody. The sensitivity standard curve of McAb secreted by (**A**) 1B6, (**B**) 2A11 and (**C**) 5G2.

**Figure 7 foods-12-01091-f007:**
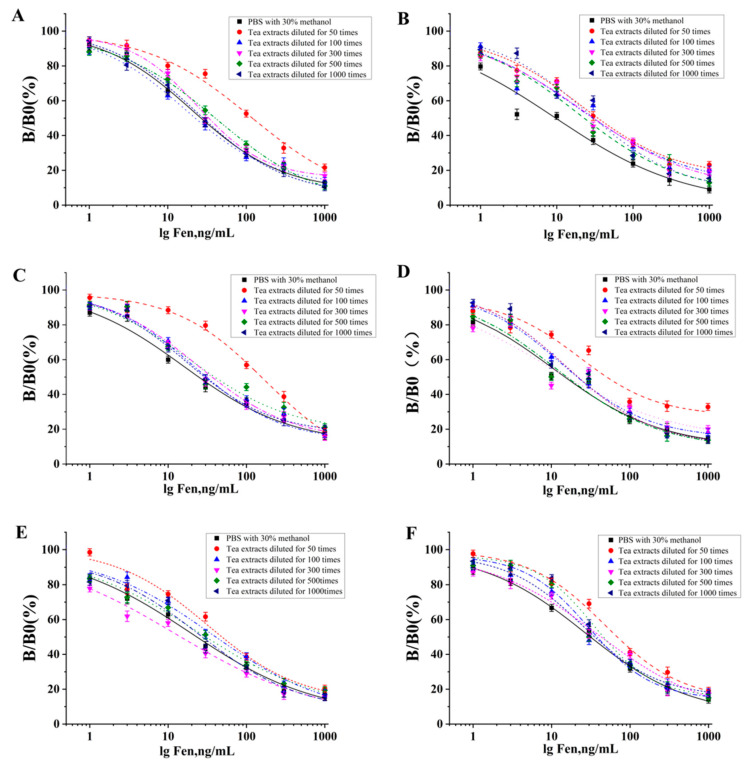
Effect of dark tea substrate on the detection of anti-Fen McAb by ic-ELISA. The influence of (**A**) Fu brick tea, (**B**) Liupao tea, (**C**) Pu‘er tea, (**D**) Qingzhuan tea, (**E**) Enshi dark tea and (**F**) selenium-enriched Enshi dark tea as a substrate on the standard curve of fenvalerate determination. Different-colored standard curves represent different concentrations of the tea substrate.

**Figure 8 foods-12-01091-f008:**
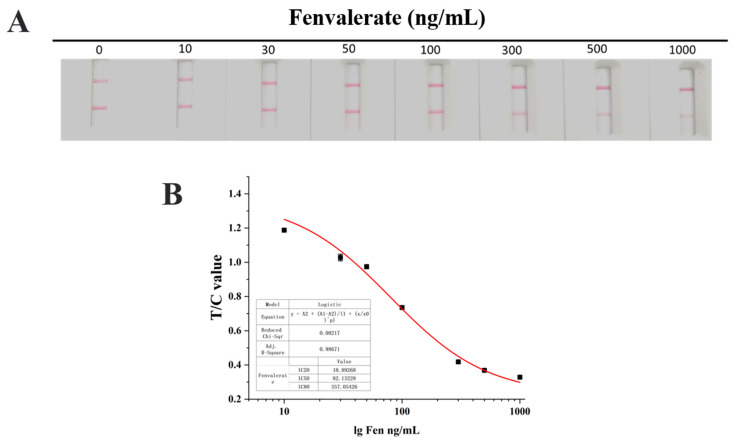
Characterization of latex microsphere immunochromatographic strips based on anti-Fen McAb. (**A**) Sensitivity of test strips. (**B**) Standard curve of the sensitivity of the test strip.

**Table 1 foods-12-01091-t001:** Comparison of the detection characteristics of the monoclonal antibodies secreted by three synthetic hybridoma cells.

Hybridoma Cells	Ka (L/mol)	IC_50_ (ng/mL)	LOD (ng/mL)	LDR (ng/mL)
1B6	2.247 × 10^9^	36.6	6.87	12.74–105.17
2A11	1.94 × 10^8^	24.29	2.6	5.93–99.45
5G2	4.52 × 10	21.65	4.96	8.55–54.85

**Table 2 foods-12-01091-t002:** Specificity identification of three McAbs against Fen.

McAb Numberi	1B6		2A11		5G2	
Pyrethroid Pesticide	IC_50_ ng/mL	CR%	IC_50_ ng/mL	CR%	IC_50_ ng/mL	CR%
Fenvalerate	36.6	100%	24.29	100	21.65	100
Deltamethrin	>10,000	<0.54	>10,000	<0.595	>10,000	<0.605
Permethrin	>10,000	<0.54	>10,000	<0.595	>10,000	<0.605
Cypermethrin	>10,000	<0.54	>10,000	<0.595	>10,000	<0.605
Bifenthrin	>10,000	<0.54	>10,000	<0.595	>10,000	<0.605
Fenpropathrin	>10,000	<0.54	>10,000	<0.595	>10,000	<0.605
Flumethrin	>10,000	<0.54	>10,000	<0.595	>10,000	<0.605
Lambda-cyhalothrin	>10,000	<0.54	>10,000	<0.595	>10,000	<0.605
D-Phenothrin	>10,000	<0.54	>10,000	<0.595	>10,000	<0.605
Cyfluthrin	>10,000	<0.54	>10,000	<0.595	>10,000	<0.605

**Table 3 foods-12-01091-t003:** Recovery of fenvalerate from spiked tea samples with McAb secreted by 1B6.

Tea Samples	Spiked (ng/mL)	Measured (ng/mL)	Recovery (%)	CV (%)
Pu‘er tea (n = 3)	25	19.2	76.8	8.2
	50	45.1	90.2	8.5
	100	99.8	99.8	7.9
Fu Brick tea (n = 3)	25	21.1	84.4	9.4
	50	48.3	96.6	8.7
	100	97.5	97.5	9.0
Qingzhuan tea (n = 3)	25	21.8	87.2	8.9
	50	44.8	89.6	9.1
	100	98.5	98.5	8.6
Liupao tea (n = 3)	25	22.2	88.8	7.7
	50	46.5	93	8.0
	100	96.7	96.7	8.4
Enshi dark tea (n = 3)	25	18.3	73.2	9.8
	50	47.2	94.4	9.2
	100	96.8	96.8	6.8
Selenium-enriched Enshi dark tea (n = 3)	25	18.6	74.4	7.8
	50	47.9	95.8	6.9
	100	99.3	99.3	6.5

## Data Availability

Not applicable.

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
