# Peer review of "Preparation, Identification and Preliminary Application of the Fenvalerate Monoclonal Antibody in Six Kinds of Dark Tea"

_foods, 2023, doi:10.3390/foods12051091_

Round 1
Reviewer 1 Report
The manuscript "Preparation, identification and preliminary application of fenvalerate monoclonal antibody in dark tea", describe a theoretical and application basis for the rapid detection of fenvalerate residues. The aim of the study it’s not easy to understand because there are many activities described. The aim should be more specific in the introduction section.
The section material and methods have few references to support the activities described.
In general, the manuscript has several references not updated. Please check all sections.
§ Moniz et al. (2005) has found that perinatal exposure to Fen resulted ...
§ Suman Gupta et al. (1996) used Fourier transform infrared (FITR)....
§ Song et al. (2011) used Fen polyclonal antibody and dc-....
§ Chinese Ministry of Agriculture stipulated that the use of Fen (2005).
§ Song et al. (2011) and McAb prepared by Jiang et al. (2010) [24,27].
Some comments are described as follow:
Title: The scientific name of dark tea should be included
1. Introduction: page 2. "Unfortunately, the large-scale use will inevitably pose a threat to the ecological environment and human beings" please include a reference.
2. Materials and methods:
2.1: Where was purchased Fenvalerate, analytical grade?
2.2: Please describe briefly the LC-MS conditions and equipment.
2.3: Please include a reference to support the synthesis
2.4: Please include references to support the synthesis of immunogens
2.5: Please include a reference to support the synthesis of coating antigen
2.6: Please include a reference to support the identification
2.7: Please include specification of BCA kit
2.7: Please include references to support the preparation and evaluation of fenvalerate.
2.8: Please include a reference to support the immunosorbent assay
3. Results:
All the references cited in this section should be included in the discussion section
The figure 6 it’s not clear to understand, to small.
The figure 7 shows similar results for all the treatment. Please reduce the number of plots.
4. Discussion
The following sentences from the manuscript "The Chinese Ministry of agriculture remove fenvalerate from the market in 1999". "There is still excessive Fe residue in tea market". The reference 26 (Wu, 2014) reported small concentrations of pyrethroids in 100 tea samples which are not representative of the total production in China in 2014. Therefore, the justification of the study should be clearer.
5. Conclusion
The provided future research directions and conclusions appears to be trivial
Author Response
REVIEWER REPORT(S):
Referee: 1
Comments and Suggestions for Authors
The manuscript "Preparation, identification and preliminary application of fenvalerate monoclonal antibody in dark tea", describe a theoretical and application basis for the rapid detection of fenvalerate residues. The aim of the study it’s not easy to understand because there are many activities described. The aim should be more specific in the introduction section.
Response: Thanks for your careful work and comments. It is our great pleasure to receive your comments. The reviewer’ comments have been responded accordingly and the revision parts have been marked in red. We have tried our best to improve the quality of our manuscript and figures.
- The section material and methods have few references to support the activities described.
In general, the manuscript has several references not updated. Please check all sections.
- Moniz et al. (2005) has found that perinatal exposure to Fen resulted ...
- Suman Gupta et al. (1996) used Fourier transform infrared (FITR)....
- Song et al. (2011) used Fen polyclonal antibody and dc-....
- Chinese Ministry of Agriculture stipulated that the use of Fen (2005).
- Song et al. (2011) and McAb prepared by Jiang et al. (2010) [24,27].
Response: Thank you for your suggestion. In terms of “Moniz et al. (2005) has found”, considering that the hazards of fenvalerate appeared in earlier studies may be more appropriate than those reported in the latest literature, we did not choose a update reference. In terms of “Suman Gupta et al. (1996)”, there were only two reference related to FTIR detection in Fen, and compared to the report of Akbal et al (2000), Suman Gupta et al. (1996) reported the detail LOD of the FTIR method in fenvalerate detection, so it was more suitable for this study. In terms of “Song et al. (2011) used Fen polyclonal antibody”, it was the updated references on fenvalerate antibody used in the detection of fenvalerate in tea. In terms of “Chinese Ministry of Agriculture stipulated that the use of Fen in1999 (2005).” Since it was the updated reference that reported the Chinese Ministry of Agriculture stipulated the use of Fen, so we used this reference. In terms of “Song et al. (2011) and McAb prepared by Jiang et al. (2010)”, since the literature search results from databases such as Web of science, Science direct, Google Scholar showed that it was the updated references that reported the preparation of polyclonal antibody and monoclonal antibody for Fen, we have used these references for several times in this study.
- Some comments are described as follow:
Title: The scientific name of dark tea should be included
Response: Thank you for your suggestion. We wonder whether it was suitable to change the tile to “Preparation, identification and preliminary application of fenvalerate monoclonal antibody in Pu 'er tea, Liupao tea, Fu Brick tea, Qingzhuan tea, Enshi dark tea, selenium-enriched Enshi dark tea” for meeting the suggestion of “scientific name of dark tea should be included”. It seems that the name of dark tea takes up too much space in the title. We suggest that the title can be revised like “Preparation, identification and preliminary application of fenvalerate monoclonal antibody in six kinds of dark tea”, and the scientific name of dark tea was added in abstract section, which were marked red in the text.
- Introduction: page 2. "Unfortunately, the large-scale use will inevitably pose a threat to the ecological environment and human beings" please include a reference.
Response: Thank you for your suggestion. The related reference was added in line 17-19, page3, as your suggestion and the revised portions were marked red in the text.
- Materials and methods:
2.1: Where was purchased Fenvalerate, analytical grade?
Response: Thank you for your careful comments. Fenvalerate standards was purchased from Dr.Ehrenstorfer GmbH (Augsburg, Germany) and it was analytical standard, the detail information was added in line 90-92, page 6 in the manuscript and the revised portions were marked red in the text.
2.2: Please describe briefly the LC-MS conditions and equipment.
Response: Thank you for your careful suggestion. Briefly, the LC-MS conditions were performed with the Acquity UPLC I-class/VION IMS QTOF (waters, Massachusetts, USA) according to Martínez with a slight modification: solvent A was ser as acetonitrile and solvent B was set as ammonium formate 50 mM, 5 % of acetonitrile, acidified at pH 3.5 by adding formic acid to performed the program: (30% B in 0-3 min, 30%-20% B in 4-8 min, 20-0% B in 8-10 min, 0% B in 10-13 min). The haptens were identified according to the liquid chromatographic peak and mass spectral molecular weight. The revised portions were marked red in line 98-104, page 3-4 in the text.
2.3: Please include a reference to support the synthesis
Response: Thank you for your suggestion. The contents of 2.3 section was a summary of 2.4 and 2.5 section, and content of this section was modified and assigned to 2.4 and 2.5 section.
2.4: Please include references to support the synthesis of immunogens
Response: Thank you for your suggestion, a reference to support the synthesis of immunogens was added in 2.4 section in line 108, page 7, and the revised portions were marked red in the text.
2.5: Please include a reference to support the synthesis of coating antigen
Response: Thank you for your suggestion, a reference to support the synthesis of immunogens was added in 2.5 section in line 143, page 8, and the revised portions were marked red in the text.
2.6: Please include a reference to support the identification
Response: Thank you for your suggestion, a reference to support the identification was added in line 155, page 9, and the revised portions were marked red in the text.
2.7: Please include specification of BCA kit
Response: Thank you for your suggestion, the specification of BCA kit BCA kit (P0010, Beyotime, China) was added to 2.6 section in line 168-169, page 9, and the revised portions were marked red in the text.
2.7: Please include references to support the preparation and evaluation of fenvalerate.
Response: Thank you for your suggestion. In our opinions, the preparation references of anti-fenvalerate monoclonal antibody were shown in 2.6 section, the method of mouse immunization was referred to xu et al [1], the methods of cell fusion and screening were performed according to classical procedure [2] and McAb in ascites was pacificated by the caprylic acid-ammonium sulfate precipitation method [3]. In terms of the evaluation of anti-fenvalerate McAb, we have reported that the affinity of McAb was determined by ic-ELISA [4], and the references for evaluating the sensitivity and specificity of anti-fenvalerate McAb were added in line 173, page, 9, and line 184, page 9, respectively, and the revised portions were marked red in the text.
2.8: Please include a reference to support the immunosorbent assay
Response: Thank you for your suggestion. Fenvalerate immunochromatographic strip was prepared according to our previous study and the related reference was added in the line 189-190, page 10, and the revised portions were marked red in the text.
5) 3. Results:
All the references cited in this section should be included in the discussion section
Response: Thank you for your careful comments. All the references cited in Results section has been included in the Discussion section, and the revised portions were marked red in the text.
The figure 6 it’s not clear to understand, to small.
Response: Thank you for your careful suggestion. Figure 6 was adjusted and became big and clear.
The figure 7 shows similar results for all the treatment. Please reduce the number of plots.
Response: Thank you for your careful suggestion. Since the six figures (A-F) correspond to the detection of anti-fenvalerate McAbs on six dark teas, rather than some parallel test, it seems to be inappropriate to remove some of the images, so we think it was suitable to be retained or put in the supplemental material.
- Discussion
The following sentences from the manuscript "The Chinese Ministry of agriculture remove fenvalerate from the market in 1999". "There is still excessive Fe residue in tea market". The reference 26 (Wu, 2014) reported small concentrations of pyrethroids in 100 tea samples which are not representative of the total production in China in 2014. Therefore, the justification of the study should be clearer.
Response: Thank you for your careful suggestion. It was indeed that 101 tea samples are not representative of the total production in China in 2014, the sentence was revised as “there are still excessive Fen residue in a research of fenvalerate detection of 101 kinds of tea samples” in line 397-398, page 18. The revised portions were marked red in the text.
- Conclusion
The provided future research directions and conclusions appears to be trivial
Response: Thank you for your careful suggestion, the future research directions were improved in the nanomaterials coupled with the antibody in the strip, the revised portions were in line 434-438, page 20, and the revised portions were marked red in the text.
Reviewer 2 Report
1. English grammar editing is necessary.
2. Describe the mathematics in further depth.
3. The discussion section has to be strengthened with some justification.
4. The sample calculation and study design should be included.
Author Response
Referee: 2
Comments and Suggestions for Authors
- English grammar editing is necessary.
Response: Thank you for your careful suggestion, our English grammar has been edited by the professional English editor, the English expression was improved, the revised portions were marked red in the text.
- Describe the mathematics in further depth.
Response: Thank you for your careful suggestion, the article has added more mathematical descriptions, for instance, the revised portions in line 391-396, page 18, were marked red in the text.
- The discussion section has to be strengthened with some justification.
Response: Thank you for your careful suggestion, the discussion section was strengthened with more justification, and the revised portions were marked red in the discussion section.
Reviewer 3 Report
I have carefully read the manuscript by Wei et al., entitled “Preparation, identification and preliminary application of fen-valerate monoclonal antibody in dark tea”, in which the authors prepared a specific and sensitive monoclonal antibody for fenvalerate to detect fenvalerate residues in dark teas which are considered hazardous to human health. The authors performed a plethora of experiments to optime the fenvalerate antigen synthesis method and obtain monoclanal antibodies for latex microsphere immunochromatographic test strip used in Fen detection. Although the topic is extensively studied, the work is of interest and experiments are well planned and performed. I have few suggestions , as reported below:
-I suggest to revise the Title because it misleads the subject of the submitted manuscript. I mean it seems like that fenvalerate is a monoclonal antibody. However, authors are preparing a monoclonal antibody to detect fenvalerate residues. Try to correct the mislead also in overall manuscript, if necessary (e.g.: the Fen monoclonal antibody is actually an anti-Fen monoclonal antibody or monoclonal antibody for Fen);
-Revise the keywords avoiding using the same words of Title;
-Check typos: e.g.: ‘IC50’ instead of ‘IC50’; the use of comas in paragraph 2.9.;
- In the Figures of the gels is ‘kDa’ instead of KDA’. Check this in overall manuscript too.
Author Response
Referee: 3
Comments and Suggestions for Authors
I have careful read the manuscript by Wei et al., entitled “Preparation, identification and preliminary application of fen-valerate monoclonal antibody in dark tea”, in which the authors prepared a specific and sensitive monoclonal antibody for fenvalerate to detect fenvalerate residues in dark teas which are considered hazardous to human health. The authors performed a plethora of experiments to optime the fenvalerate antigen synthesis method and obtain monoclanal antibodies for latex microsphere immunochromatographic test strip used in Fen detection. Although the topic is extensively studied, the work is of interest and experiments are well planned and performed. I have few suggestions, as reported below:
Response: Thanks for your careful work and comments. It is our great pleasure to receive your comments. The reviewer’ comments have been responded accordingly and the revision parts have been marked in red. We have tried our best to improve the quality of our manuscript and figures.
-I suggest to revise the Title because it misleads the subject of the submitted manuscript. I mean it seems like that fenvalerate is a monoclonal antibody. However, authors are preparing a monoclonal antibody to detect fenvalerate residues. Try to correct the mislead also in overall manuscript, if necessary (e.g.: the Fen monoclonal antibody is actually an anti-Fen monoclonal antibody or monoclonal antibody for Fen);
Response: Thanks for your suggestion, the expression has been corrected in detail, “the monoclonal antibody of fenvalerate” was revised as “anti-Fen monoclonal antibody”. The revised portions were marked red in the text.
-Revise the keywords avoiding using the same words of Title;
Response: Thanks for your suggestion, the keywords were revised according to your suggestion.
-Check typos: e.g.: ‘IC50’ instead of ‘IC50’; the use of comas in paragraph 2.9.;
Response: Thanks for your careful suggestion, the ‘IC50’ has been revised as ‘IC50’ in paragraph 2.9 according to your suggestion.
- In the Figures of the gels is ‘kDa’ instead of KDA’. Check this in overall manuscript too.
Response: Thanks for your careful suggestion, the ‘KDA’ has been revised as ‘kDa’ in overall manuscript according to your suggestion.
Round 2
Reviewer 1 Report
Dear Authors
Congratulations,
The manuscript is suitable for publication,
Kind Regards
Reviewer 2 Report
The authors improved the manuscript and it should be accepted in present form.